# CXC Chemokine Signaling in Progression of Epithelial Ovarian Cancer: Theranostic Perspectives

**DOI:** 10.3390/ijms23052642

**Published:** 2022-02-27

**Authors:** Xinxin Huang, Juncheng Hao, Yan Qin Tan, Tao Zhu, Vijay Pandey, Peter E. Lobie

**Affiliations:** 1Tsinghua-Berkeley Shenzhen Institute, Tsinghua Shenzhen International Graduate School, Tsinghua University, Shenzhen 518055, China; hxx19@mails.tsinghua.edu.cn (X.H.); haojc20@mails.tsinghua.edu.cn (J.H.); esthertan@sz.tsinghua.edu.cn (Y.Q.T.); 2Institute of Biopharmaceutical and Health Engineering, Tsinghua Shenzhen International Graduate School, Tsinghua University, Shenzhen 518055, China; 3Hefei National Laboratory for Physical Sciences at Microscale and School of Life Sciences, University of Science and Technology of China, Hefei 230000, China; zhut@ustc.edu.cn; 4The CAS Key Laboratory of Innate Immunity and Chronic Disease, School of Life Sciences, University of Science and Technology of China, Hefei 230000, China; 5Shenzhen Bay Laboratory, Shenzhen 518055, China

**Keywords:** ovarian cancer, CXC chemokine ligand (CXCL), CXC chemokine receptor (CXCR), tumor microenvironment (TME)

## Abstract

Patients with epithelial ovarian cancer (EOC) are often diagnosed at an advanced stage due to nonspecific symptoms and ineffective screening approaches. Although chemotherapy has been available and widely used for the treatment of advanced EOC, the overall prognosis remains dismal. As part of the intrinsic defense mechanisms against cancer development and progression, immune cells are recruited into the tumor microenvironment (TME), and this process is directed by the interactions between different chemokines and their receptors. In this review, the functional significance of CXC chemokine ligands/chemokine receptors (CXCL/CXCR) and their roles in modulating EOC progression are summarized. The status and prospects of CXCR/CXCL-based theranostic strategies in EOC management are also discussed.

## 1. Introduction

Ovarian cancer (OC) is the third most prevalent and second most lethal gynecological malignancy worldwide, with an estimated 6.6% incidence and 4.2% mortality raterespectively [1]. OC is classified into epithelial, germ cell, sex cord-stromal, and other unspecified subtypes, with epithelial ovarian cancer (EOC) being the most common, accounting for approximately 90% of all cases [2]. EOC can be further stratified into high-grade serous (52%), endometrioid (10%), clear cell (6%), and mucinous (6%) carcinomas among others [3]. In contrast to non-epithelial OC, EOC is diagnosed in about 65% of cases at advanced stages (Stages III/IV) [2]. Due to the lack of specific symptoms and effective screening methods, most patients with EOC fail to receive timely treatment [4,5,6]. Recurrence of the disease represents another clinical challenge in EOC and results in 5-year overall survival (OS) rate of 48% [1]. The extrinsic contributing factors in OC have not been well delineated, but individuals with a family history of ovarian or breast cancer are at higher risk, and approximately 40% of these individuals have *BRCA1* and *BRCA2* mutations [7].

Chemokines are a subset of immune cell-attracting and modulating cytokines which are integral to the inflammatory response and immune homeostasis [8]. Evidence [9,10,11,12,13] has accrued indicating that those chemokines participate in the regulation of cancer cell proliferation, invasion, angiogenesis, and therapy resistance, primarily through recruitment of immune cells and modulation of the tumor microenvironment (TME). The present review is dedicated to elaborating the roles of the CXC chemokine family in OC progression and summarizes the status quo of CXC chemokine-based theranostic applications in OC management.

Chemokines are grouped into four distinct subfamilies—C, CC, CXC, and CX3C—based on the position of the first two cysteines (C) in the primary sequence, where ‘X’ stands for a non-conserved amino acid. Given the presence or absence of a three-amino-acid-motif ELR (glutamic acid–leucine–arginine) preceding the CXC sequence, CXC chemokines are further stratified into ELR^+^ and ELR^−^ subtypes [9]. Classical CXC chemokine receptors are coupled with trimeric G-proteins (G_αβγ_), and the intracellular effects of their activation depend on the functions of four types of Gα subunits [10,11,12]. Gα_s_ and Gα_i_ regulate cAMP levels by stimulating and inhibiting adenylate cyclase (AC), respectively; cAMP can further activate protein kinase PKA. Gα_q_ stimulates phosphatidylinositol (PI)-specific phospholipase C (PLC) and the hydrolysis of phosphatidylinositol biphosphate (PIP_2_), generating twosecond messengers, diacylglycerol (DG) and inositol 1,4,5-triphosphate (IP_3_); DG and IP_3_ can activate protein kinase PKC and stimulate intracellular calcium release. G_βγ_ complex has also been reported to trigger PLC activation. Gα_12_ exerts its functions primarily through other small monomeric G-proteins. Further downstream of chemokine receptor pathways, phosphatidylinositol 3-kinase (PI3K), mitogen-activated protein kinase (MAPK), signal transducer and activator of transcription (STAT), and nuclear factor κB (NF-κB) cascades represent four major events promoting cell survival and chemotaxis [10,11,12]. 

The pairing of CXC chemokines and chemokine receptors are presented in Table 1. ELR^+^ CXC chemokines share a common receptor, CXCR2, except for CXCL8 which can also bind to CXCR1. CXCL1, CXCL2, and CXCL3 are three variants of growth regulated oncogene (GRO). CXCL14 is a newly identified ELR^−^ chemokine, and no receptor has yet been confirmed for CXCL14; however, it has been shown to exhibit binding affinity for CXCR4 and thereby inhibit the CXCL12/CXCR4 axis [8]. In general, ELR^+^ CXC chemokines are potent stimulators of angiogenesis via activation of CXCR2 on endothelial cells; in contrast, CXCR3-binding ELR^−^ CXC chemokines exert inhibitory effects on angiogenesis [8]. The biological functions of CXC chemokines in EOC progression are further discussed below.

## 2. Functions of CXC Chemokines and Their Receptors in EOC

Increased expression of CXCL1/8/10/12 have been observed during ovulation and luteal regression as a normal part of the menstrual cycle [13,14,15,16,17]. However, continued high expression of CXC chemokines recruits inflammatory cells and creates a persistent inflammatory microenvironment favorable for cancer development [18,19,20]. As shown in Table 1, some CXC chemokines promote cancer progression, while others inhibit it. 

**Table 1 ijms-23-02642-t001:** Cancer promoting or inhibiting functions of CXC chemokines and their receptors in EOC.

Family	Chemokines	Alternative name	Receptors	Functions	References
• CXC chemokines promoting cancer progression					
ELR^+^	CXCL1	Growth related oncogene (GRO)-α	CXCR2	Promotes cancer cell proliferation and invasion Recruits neutrophils	[21,22]
	CXCL2	GRO-β	CXCR2	Promotes cancer cell proliferatoin and angiogenesis	[21,23]
	CXCL5	Epithelial neutrophil-activating peptide (ENA)-78	CXCR2	Promotes cancer cell metastasis and angiogenesis	[24]
	CXCL8	Interleukin (IL)-8	CXCR1, CXCR2	Promotes cancer cell proliferation, metastasis, and angiogenesis Recruits neutrophils	[25,26,27,28]
	CXCL11	IFN-inducible T-cell α-chemoattractant (I-TAG)	CXCR3	Promotes cancer cell proliferation and invasion	[29]
	CXCL12	Stromal cell-derived factor 1 (SDF1)-α	CXCR4, CXCR7	Promotes cancer cell proliferation, metastasis, and angiogenesis Promotes macrophage polarization towards M2	[30,31,32,33,34,35]
ELR^−^	CXCL14	Breast and kidney-expressed chemokine (BRAK)	-	Promotes cancer cell proliferation	[36]
	CXCL16	Scavenger receptor for phosphatidylserine and ox-LDL (SR-POX)	CXCR6	Promotes cancer cell invasion and metastasis	[37,38]
• CXC chemokines with tumor-suppressive functions					
ELR^+^	CXCL3	GRO-γ	CXCR2	Suppresses stromal cells and cancer formation	[39]
ELR^−^	CXCL4	Platelet factor (PF)-4	CXCR3	Inhibits angiogenesis	[40,41]
	CXCL9	Monokine induced by interferon-γ (MIG)	CXCR3	Recruits T cells Promotes macrophage polarization towards M1	[42]
	CXCL10	Interferon-γ-inducible protein (IP)-10	CXCR3	Recruits T cells and NK cells	[43]
	CXCL11	I-TAG	CXCR3, CXCR7	Recruits T cells and increases immune cell infiltration	[44]
	CXCL13	B-cell attracting chemokine (BCA)-1	CXCR3, CXCR5	Activates tumor-infiltrating immune cells	[45]
	CXCL16	SR-POX	CXCR6	Recruits T cells	[46]

### 2.1. Oncogenic Functions

Chemokine ligands and their receptors have been observed to be associated with cancer cell proliferation and migration, as well as angiogenesis and immune suppression in EOC [47].

#### 2.1.1. Proliferation and Angiogenesis

By activating a variety of signaling pathways (such as MAPK, NF-κB, and STAT3 pathways), CXCR2 and its ligands (CXCL1/2/8) regulate expression of cell cycle-associated proteins (Cyclin A/B1/D1, CDK2, CDK6), apoptotic proteins (TP53, PUMA, BCL-xS), and angiogenic factors [23,26,27]. The elevated levels of CXCL1/2/8 signal through CXCR2 in a paracrine manner topromote endothelial cell tube generation and also to promote EOC cell proliferation in an autocrine manner [48]. Consistently, depletion of CXCL1/8 or CXCR2 expression with siRNA significantly reduces the proliferation of EOC cells [21,23,25]. In addition, CXCL1 induces fibroblast senescence via the TP53 pathway, and CXCL1 expression can be enhanced by RAS to promote EOC cancer development by promoting fibroblast senescence [49]. Enhanced expression of CXCL1 can transactivate EGFR through CXCR2, leading to the activation of the MAPK signaling pathway and promotion of EOC cell proliferation [21].

It has been reported that aberrant expression of CXCL12/SDF-1α and its receptor CXCR4 in EOC tissues promotes EOC cell invasion and angiogenesis [50]. CXCL12 attracts plasmacytoid dendritic cells (pDCs) into the TME and induces pDCs to produce tumor necrosis factor (TNF)-α and CXCL8 to promote angiogenesis [51,52]. Under the hypoxic conditions prevalent in cancer [53], hypoxia-inducible factor (HIF)-1α selectively enhances CXCR4 expression and chemotactic responses to CXCL12—including EOC cells, TAMs, and endothelial cells—thus inducing angiogenesis [54]. Depletion of CXCR4 with siRNA inhibits cell proliferation, leading to G1/S arrest and increased apoptosis and chemosensitivity [55]. Therefore, increased expression of the CXCR4/CXCL12 axis promotes cancer progression; hence patients with higher CXCR4 expression exhibit chemoresistance and reduced progression-free survival (PFS) and OS [55]. In addition, increased CXCL11 expression in cancer-associated fibroblasts (CAFs) promotes the proliferation and migration of EOC cells via CXCR3 [29]. Furthermore, high CXCL14 expression in patients with advanced EOC increases phosphorylation of STAT3 and promotes the proliferation of EOC cells [36].

#### 2.1.2. Metastasis

It has been reported that EOC cells expressing CXCR2 exhibit enhanced NF-κB activation via EGFR trans-activated AKT signaling [21,22], inducing the secretion of CXCL1/2 and promoting EOC peritoneal dissemination leading to higher mortality [24,56,57]. In addition, through binding to CXCR1/2, CXCL8 activates the WNT/β-catenin pathway to promote epithelial–mesenchymal transition (EMT) and EOC metastasis [28]. 

CXCR4 is expressed on normal hematopoietic stem cells (HSCs) and non-hematopoietic tissue-committed stem/progenitor cells (TCSCs). Hence, HSCs and TCSCs are attracted by CXCL12, which is the ligand of CXCR4 [30]. Thus, high CXCR4 expression in cancer stem cells (CSCs) will drive metastasis to tissues with high CXCL12 expression, such as the liver, lymph nodes, and bones [30]. It has been reported that SKOV3 cells overexpressing CXCR4 exhibit an increased ability to proliferate and migrate in response to CXCL12 compared to normal SKOV3 cells. In addition, SKOV3 cells co-cultured with human peritoneal mesothelial cells (HPMC) which secrete CXCL12 results in higher migration and invasion when compared to SKOV3 cells alone [31]. CXCL12 decreases the expression of the tumor-suppressor ARHGAP10 through VEGF signaling and promotes EOC cell invasion [32]. High levels of WNT5A (noncanonical WNT pathway) have been observed in ascites due to EOC which promotes adhesion of EOC cells to PMCs as well as EOC migration and invasion. WNT5A enhances CXCR4/ CXCL12 axis signaling and promotes EOC metastasis through activation of protein kinase C [33,58]. 

CXCR7, another receptor for CXCL12, was also observed to be highly expressed in EOC. CXCR7/CXCL12 enhances cell invasion through p38 MAPK dependent enhanced matrix metallopeptidase 9 (MMP-9) expression [59]. It was observed that estrogen receptor α (ERα) enhances CXCR7 expression in EOC cells, and one of its ligands, CXCL11, is also upregulated by estrogen to promote EMT and metastasis of EOC cells [34]. High expression of CXCL11 with its other receptor, CXCR3, has also been observed to promote EOC metastasis. In addition, cancer cell-derived lymphotoxin can induce CXCL11 expression in stromal fibroblasts through the lymphotoxin receptor (LTBR)-NF-κB signaling pathway, thus contributing to EOC lymph node metastasis [29]. 

The CXCR6/CXCL16 axis is also involved in the peritoneal metastasis of EOC [37]. It has been reported that in co-culture of macrophages and EOC cells (SKOV3), with added TNF-α to mimic the TME, the expression of CXCL16 and CXCR6 was increased in SKOV3 cells. In addition, by enhancing the CXCR6/CXCL16 pathway, TAMs increase the level of phosphorylated AKT and promote EOC cell migration and invasion [38].

#### 2.1.3. Immunosuppression

CXC chemokines and chemokine receptors are important for recruiting immune cells by targeted migration from the circulatory system to sites of inflammation. Such immune cell recruitment—including TAMs, myeloid-derived suppressor cells (MDSCs), dendritic cells (DCs), and neutrophils— drive immune suppression in EOC [60].

TAMs are the most numerous infiltrating immune-associated cells regulating cancer progression and distant metastasis [61]. Macrophages are classified into a classically activated (M1) phenotype and an alternatively activated (M2) phenotype. It is generally accepted that TAMs belong to pro-angiogenic and immunosuppressive M2 phenotypes, which may induce a CSC-like transformation of cancer cells [35]. Analysis of tumor samples from EOC patients revealed that metastatic sites exhibited more total TAM and a lower M1/M2 ratio than primary tumors, while larger vessels in the tumor were also observed to increase the extent of M2 infiltration [62]. 

Hypoxia is a hallmark of cancer that affects macrophage distribution and function. EOC cells, together with the accumulated TAMs, promote angiogenesis in hypoxic environments [63,64]. It has been reported that HIF-1α/2α induces miRNA delivery to macrophages from EOC cell-derived exosomes in a hypoxic environment to induce macrophage polarization toward M2, thereby promoting EOC cell proliferation and migration [65]. HIF-2α increases CXCR4 expression in macrophages [66]. In addition, macrophages are attracted to CXCL12 secreted by CAFs and EOC cells, causing them to exhibit an M2 macrophage phenotype [35], thereby promoting cancer cell proliferation.

Macrophage migration inhibitory factor (MIF) secreted by CSCs inhibits anti-cancer immune responses via the CXCR2 pathway, increasing MDSC arginase-1 (ARG-1) expression and inhibiting CD8+T cell migration [67]. It was observed that precursor plasmacytoid dendritic cells (preDC2) expressing CXCR4 are attracted to the TME by EOC cells expressing CXCL12, resulting in poor T-cell activation and immunosuppression. Notably, CXCL12 secreted by cancer cells also protects preDC2 from macrophage-mediated IL-10-induced apoptosis [68]. In addition, CXCL1/5 promotes neutrophil recruitment [69] and IL-17A produced by neutrophils stimulates EOC stromal cells to secrete CXCL1, thereby attracting more neutrophils to form an aberrant immune environment that promotes cancer progression and suppresses anti-tumor immune responses [19]. 

### 2.2. Tumor-Suppressor Functions

CXCL9, CXCL10, and CXCL11 bind to CXCR3 and regulate lymphocyte chemotaxis to mediate recruitment of tumor-infiltrating lymphocytes (TIL) for tumor-suppressor activity [70,71]. By genetic analysis of multiple EOC tissues, it was reported that increased expression of CXCL9 and CXCL10 correlates positively with increased infiltration of CD8+T cells and is associated with reduced metastasis and improved OS in EOC patients [72,73]. In EOC ascites and tumor models, in which macrophages overexpress IKK2, there is a sustained shift of macrophage populations from M2 to an anti-tumor M1 phenotype. The activation of the NF-κB signaling pathway in macrophages increases CXCL9 expression and promotes an anti-tumor TAM phenotype (M1), thereby increasing cytotoxic T cell infiltration and limiting EOC progression [42]. Decreased expression of CXCL9 and CXCL10 renders EOC cells resistant to anti-tumor T-cell infiltration and eventually leads to immune escape [74,75]. Increased CXCL10 expression promotes NK cell and CD8+T cell infiltration and inhibits EOC progression [43,76,77]. Increased expression of cyclooxygenase-2 (COX-2) renders EOC cells resistant to chemotherapy and radiotherapy [44]. The COX-2 inhibitor, indomethacin, increases the expression of CXCL9/10 in EOC cells; a phenomenon also observed in breast cancer (BC) [78]. In addition, it has been reported that inhibition of Toll-like receptors promotes the expression of CXCL9/10/11 in mouse models of colorectal cancer (CRC) and BC [79].

CXCL11 also promotes T cell chemotaxis and plays a role in anti-tumor immunity [44]. Hence, high mRNA expression levels of CXCL10 and CXCL11 are positively associated with prolonged OS in EOC patients [80]. Strategies to directly increase the levels of CXCR3 ligands have been described in preclinical models in Table 2. These include recombinant CXCL10 protein [76], plasmids expressing CXCL10 [81], and an increase in CXCL9/10 expression due to the effect of drug treatment on signaling pathways [44]. Furthermore, CXCL9 has been proposed as a therapeutic for EOC due to its ability to inhibit angiogenesis and promote anti-tumor immunity [82]. 

Most of the ligands of CXCR3 in the ELR^−^ family possess tumor-suppressor functions. In addition to CXCL9, CXCL10, and CXCL11, CXCL4 and CXCL13 also produce increased immune responses and exhibit anti-angiogenic functions. By direct interaction with VEGF and fibroblast growth factor-2 (FGF-2), CXCL4 inhibits the effect of these angiogenic factors and retards cancer growth [40,83]. Additionally, the expression of CXCL4 and its receptor CXCR3 are observed to be downregulated in clear cell OC, possibly due to TAM inhibition, creating a microenvironment suitable for cancer cell growth [41]. Furthermore, high CXCL13 expression in EOC exhibits increased infiltration of activated and CXCR5-expressing CD8+T cells, delaying cancer growth and increasing patient survival [45]. It was also reported that CD8+T cells which express CXCR6 are recruited by CXCL16 secreted by cancer cells, hence, enhanced CXCR6 expression in CD8+T cells enhances cancer T cell infiltration and limits EOC progression [46,84]. Additionally, a chemokine tumor-inducing factor (Tif), a possible hamster homolog of human GROγ (CXCL3), has been identified in a mouse ovarian xenograft model and suppresses EOC growth by inhibiting stromal fibroblast proliferation [39]. 

In summary, CXC chemokines exert a major impact on immune cell recruitment in the TME. Therapies that inhibit CXCL/CXCR signaling may be easier to develop than methods that increase CXCL/CXCR expression in cancer. However, further investigation on the expression of CXCL/CXCR in EOC and their relationship with immune cells may provide further opportunities for CXCL/CXCR-based precision therapies. In Figure 1, a summary ofthe oncogenic and tumor-suppressive functions of CXCLs/CXCRs in EOC is provided. 

## 3. Potential CXCL/CXCR-Based Theranostic Strategies

The standard treatment option for advanced EOC has been surgical resection followed by chemotherapy, typically platinum-based agents and paclitaxel [85]. Although this approach has led to improved survival rates, many patients experience recurrence of the disease due to the development of chemoresistance [86,87,88]. One of the reasons for chemotherapy resistance is drug-induced activation of immunosuppressive and oncogenic mechanisms in the TME, which heavily relies on a variety of chemokines [89,90,91,92]. CXC chemokines, as a major chemokine subfamily, serve as promising biomarkers and druggable targets in EOC.

### 3.1. CXCL/CXCR-Associated Targeted Pre-Clinical Strategies

As previously summarized, inhibition of increased CXCR expression in EOC enhances immunosuppression and reduces cancer progression. Specific targeting of CXCRs with inhibitors is presented in Table 2. Le Naour et al. demonstrated that the CXCR1/2 inhibitor, AS-62401, reduces the oncogenic functions of CXCR1/2 ligands and increases the sensitivity of EOC cells to carboplatin when used in combination [93]. Another CXCR2 inhibitor, SB225002, was observed in multiple experiments to inhibit EOC cell proliferation and metastasis as well as angiogenesis [21,94,95]. In addition to CXCR2 inhibitors, the CXCR4 inhibitor AMD3100 can inhibit metastasis [32,96,97] and increase the infiltration of TILs [98]. In addition, OVV-CXCR4-A-Fc, a viral construct expressing a CXCR4 antagonist combined with the Fc fragment of mouse IgG2a possesses the same function [99]. 

TILs protect against cancer development/progression, and chimeric antigen receptor-T cell (CAR-T) therapy is a potential treatment approach for solid tumors [100]. As previously mentioned, EOC cells and tumor-associated immune cells secrete multiple CXCR2 ligands, such as CXCL1/2/8. However, CXCR2 is not expressed in T cells, and modifying T cells to express CXCR2 is beneficial in directing T cell migration towards cancer and infiltrating the cancer mass to enhance the anti-tumor response [101]. NCT01740557 investigated the role of CXCR2-transduced autologous TILs for the treatment of metastatic malignant melanoma (MM). It is important to note that CXCL8 is also expressed in tissues with an inflammatory response, so the absence of autoimmune disease or other chronic inflammatory conditions should be determined before utilizing this method [102]. Clinical trials have been conducted to determine the effectiveness of CXCR5-expressing anti-EGFR CAR-T cells in patients with advanced EGFR positive NSCLC (NCT05060796, NCT04153799). Additionally, CXCR6 can be used to promote resident memory T cell-mediated immunosurveillance and control EOC progression [46]. Furthermore, CXCR1-expressing NK cells are attracted to the TME by CXCL8 secreted by EOC cells, thus enhancing NK cell infiltration and the anti-tumor response [43,103]. 

Current therapeutic developments in the CXCL/CXCR axis are mainly focused on targeting receptors. However, studies have also observed that targeting CXCL associated pathways shows promising therapeutic effects, with specific targeting strategies shown in Table 2. The modulation of CXCL function by targeting the CXCL/CXCR axis-associated signaling pathways has been widely studied. EGFR inhibitor (PD153035) reduces the proliferation of EOC cells promoted by CXCL1 through the MAPK signaling pathway [21]. IKK inhibitors (TPCA-1, IKK16, Bay65-1942, and Bay 117085, HDI) reduce CXCL8 expression to inhibit EOC growth and angiogenesis, and TPCA-1 and IKK16 also abrogates the proliferation of EOC cells by reducing CXCL1/2 expression [48]. PAI-1 inhibitor (Tiplaxtinin) prevents the formation of CAMs and alleviates peritoneal metastases in patients with EOC [24]. COX2 inhibitor (Celecoxib) prevents CXCL12 from attracting MDSCs and reduces immunosuppressive effects [104]. STAT3 inhibitor III (CAS 19983-44-9) limits CXCL14-induced proliferation of EOC cells [36].

CXCLs with tumor-suppressive activity may also be exploited in anti-cancer treatment. The COX inhibitor (Indomethacin) inhibits EOC progression by recruiting TILs through increasing CXCL9/10/13 expression. The combination of a CDK4/6 inhibitor (Abemaciclib) and a PD1 inhibitor decreases OC growth by increasing secretion of CXCL10/13 which then increases infiltration of CD8+T cells and B cells [44,104,105]. CXCL10-mucin-GPI, made by fusing CXCL10 to mucin structural domains and glycosylphosphatidylinositol (GPI)-anchored signaling sequences, enhances the recruitment of NK cells and increases immune cell infiltration [76]. This approach could be extended to a wider variety of immune cells in the future, allowing targeted recruitment of cells in a variety of settings to promote anti-tumor immune responses [76].

### 3.2. CXCL/CXCR-Associated Targeted Clinical Strategies

The current clinical trials targeting CXCLs/CXCRs are listed in Table 2. There are a limited number of ongoing clinical trials that are targeting CXCL/CXCR in EOC. However, the CXCL/CXCR targeting strategies used in other solid cancer types will indicate potential utility in EOC. 

The front-line therapy for EOC treatment has been carboplatin and paclitaxel [85]. Due to the complex composition and interaction between EOC and the TME, single-drug treatments can easily lead to the development of drug resistance. Therefore, combination therapies may provide a better strategy with improved efficacy in the treatment of EOC [106]. In the clinical trials listed in Table 2, there are four clinical trials for the combination of CXCR2 inhibitors and paclitaxel (NCT02583477, NCT02370238, NCT02001974, NCT01861054). Three of these trials used Reparicin (CXCR1/2 inhibitor) in combination with paclitaxel to treat patients with HER-2 negative metastatic breast cancer (MBC) and triple-negative breast cancer (TNBC), and the results of these clinical trials showed that the combination with Reparicin appeared to be safe and well-tolerated in patients (NCT02370238, NCT02001974, NCT01861054) [107,108,109]. However, the results of the clinical trial using another CXCR2 inhibitor, AZD5069, showed that all 23 patients in the trial experienced adverse events (AE), and 74% had AEs related to the treatment drug, so the trial was not completed (NCT02583477). 

Inhibition of programmed death-1 (PD-1) is one cancer immunotherapy approach [110]. The main PD-1 inhibitors used in the clinical trial studies listed herein were Nivolumab and Pembrolizumab. However, the use of PD-1 inhibitors alone suffers from a lack of sustained clinical benefit [111] and PD-1 inhibitors have therefore been combined with CXCL/CXCR inhibitors to improve the response rate [112]. The combination of Nivolumab and HuMax-IL8 (CXCL8 inhibitor) was tested in multiple cancers, including renal cell carcinoma (RCC), malignant melanoma, NSCLC, hepatocellular carcinoma (HCC), prostate carcinoma (PC), and squamous cell carcinoma of head and neck (SCCHN) (NCT04572451, NCT04123379, NCT03689699, NCT04848116). HuMax-IL8 is effective in the treatment of solid cancers with a low rate of AEs and maintained stable disease in 73% of patients with metastatic or unresectable solid cancers [113]. Another clinical trial investigating the combination of Navarixin (CXCR1/2 inhibitor) and pembrolizumab for the treatment of NSCLC, castration-resistant prostate cancer (CRPC), and microsatellite stable (MSS) CRC is ongoing (NCT03473925). Another CXCR1/2 inhibitor, SX-682, an orally available small molecule, blocks the entry of MDSCs into cancer and can promote an immune response to cancer. It is currently being trialed in combination with Pembrolizumab, in an attempt to improve the response rate to immunotherapy (NCT03161431). Olaptesed (NOX-A12) targets CXCL12, which alone or in combination with Pembrolizumab is a potential therapy for CRC and PDAC (NCT03168139). In addition, Pentixafor (CXCR4 inhibitor) is also being explored in combination with another PD-1 inhibitor (Cemiplimab) for the treatment of PDAC (NCT04177810).

Some clinical trials have exploited the anti-tumor functions of specific chemokines to treat human cancer. Sitagliptin increases CXCL10 expression by inhibiting dipeptidyl peptidase IV (DPPIV) and regulates lymphocyte transport to hepatocellular carcinoma (HCC) to promote regression (NCT02650427). NG-641 is an oncolytic transgene expressing adenoviral vector, increasing the expression of CXCL9, CXCL10, and IFN-α for the treatment of patients with metastatic or advanced epithelial tumors (NCT04053283).

**Table 2 ijms-23-02642-t002:** CXCL/CXCR-targeted strategies in preclinical and clinical studies.

Anticancer Agent	Target	Cancer Type	Trial Phase	Outcomes
CAR-NK-CXCR1	CXCR1	OC	Preclinical	CAR-NK cells express CXCR1 and are induced to migrate toward cancer cells by secreted CXCL8, exerting tumor- suppressive effects [103].
AS-62401	CXCR1/2	OC	Preclinical	Sensitizes EOC cells to carboplatin and helps to reduce the development of cellular resistance [93].
SB225002	CXCR2	OC	Preclinical	In combination with sorafenib blocked the CXCL8-activated VEGF cytokine pathway and improved the outcome of anti-angiogenic therapy [94]. Inhibition of CXCR2 significantly attenuates NF-κB signaling and reduces oncogenicity and metastasis [95].
AMD3100	CXCR4	OC	Preclinical	Inhibited CXCL12-CXCR4 enhancement of ARHGAP10 expression, thereby suppressing EOC cell invasion [32]. In combination with taxol, significantly reduced the proliferation of human and mouse EOC cells [96]. CXCR4 inhibition increased T cell-mediated anti-tumor immune response and slowed EOC progression [98].
OVV-CXCR4-A-Fc	CXCR4	OC	Preclinical	Inhibition of peritoneal spread of EOC and improved survival [99].
PD153035	CXCL1	OC	Preclinical	PD153035 (EGFR inhibitor) inhibits CXCL1-induced ERK1/2 phosphorylation and suppresses EOC cell proliferation [21].
TPCA-1, IKK 16, Bay 65-1942	CXCL1/2/8	OC	Preclinical	Inhibition of IKK suppressed EOC growth and angiogenesis by downregulating CXCL1/2/8 expression [48].
Tiplaxtinin	CXCL5/8	OC	Preclinical	By inhibiting PAI-1, tiplaxtinin downregulates CXCL5/8 and reduces peritoneal metastasis in EOC [24].
Bay 117085	CXCL8	OC	Preclinical	The combination of BAY-117085 and bortezomib reduced CXCL8 and NF-κB (p65) expression, reducing EOC growth and increasing the effectiveness of EOC treatment [114].
Indomethacin *	CXCL9/10	OC	Preclinical	The inhibition of COX by indomethacin increases CXCL9/10 expression which increases TILs infiltration and inhibits EOC growth [44].
CXCL10-GPI Anchored Fusion Protein *	CXCL10	OC	Preclinical	Endothelial cells incubated with CXCL10-mucin-GPI activated the CXCR3 receptor on lymphocytes and recruited NK cells in vitro [76].
HPEI+pVITRO-CXCL10 *	CXCL10	OC	Preclinical	Delivery of pVITRO-CXCL10 to EOC cells using HPEI nanogels inhibits the growth of EOC [81].
Celecoxib	CXCL12	OC	Preclinical	COX2 inhibition blocks CXCL12 production in EOC and the ability to attract MDSC, reducing the immunosuppressive effect of cancer [104].
Stattic	CXCL14	OC	Preclinical	Increased expression of CXCL14 enhances the phosphorylation of STAT3 in EOC cells activating this pathway and promoting cell proliferation [36].
Navarixin	CXCR1/2	NSCLC, PC, CRC	Phase 2	Completed, no results posted (NCT03473925)
Reparixin	CXCR1/2	BC	Phase 2	Reparixin inhibits CXCR1/2 expression and reduces BC metastases. The combination with paclitaxel is safe and tolerable for patients. (NCT02370238 [107], NCT02001974)
Pentixafor	CXCR4	PDAC	Phase 2	Recruiting (NCT04177810)
CXCR5-EGFR CAR-T *	CXCR5	NSCLC	Phase 1	Recruiting (NCT05060796) Recruiting (NCT04153799)
HuMax-IL8	CXCL8	MM, RCC	Phase 1	Recruiting (NCT04572451) Recruiting (NCT04848116)
		SCCHN, NSCLC, HCC, PC	Phase 2	Recruiting (NCT04848116) Recruiting (NCT04123379) Recruiting (NCT03689699)
NG-641 *	CXCL9/10	Epithelial Tumor	Phase 1	Recruiting (NCT04053283)
Sitagliptin *	CXCL10	HCC	Phase 1	Completed, no results posted (NCT02650427)
Olaptesed	CXCL12	CRC, PDAC	Phase 1, 2	Completed, no results posted (NCT03168139)

* Therapies based on CXCL anti-tumor effects. OC, ovarian cancer; MM, multiple myeloma; NSCLC, non-small cell lung cancer; PC, prostate cancer; CRC, colorectal cancer; PDAC, pancreatic ductal adenocarcinoma; BC, breast cancer; RCC, renal cell carcinoma; SCCHN, squamous cell carcinoma of the head and neck; HCC, hepatocellular carcinoma; HPEI, heparin-polyethyleneimine.

### 3.3. CXCL/CXCR Used in Other Strategies

Early detection and evaluation of EOC is critical as it renders treatment more effective [115]. One study has reported elevated serum levels of CXCL16 triggered by A disintegrin and metalloproteinase (ADAM) as a potential marker indicative of highly metastatic OC and poor outcomes in patients [116]. Another study revealed a positive correlation of combined IFNG, CD30, CXCL13, and PRF1 with increased immune cell infiltration, which predicted anti-tumor immunity and prolonged OS in patients with EOC [117]. CXCL12 has been associated with progression in multiple types of cancer [118,119,120]; however, its prognostic potential in EOC remains controversial [121,122,123]. The fact that CXCL12 is constitutively expressed in normal ovarian surface epithelium (OSE) and fallopian tubes partially accounts for this ambiguity, and further investigation is needed. Pro-angiogenic CXCL8 is associated with a poor prognosis of EOC in preclinical studies [25,57,124]. Clinical trials (NCT00798655, NCT04810585, NCT02329639) are examining the potential of CXCL8 as a biomarker for therapy response and prognosis in a panel of different cancers; which would hopefully include EOC in the near future.

The application of nanoparticles in the treatment of cancer has been widely explored in recent years. The combination of nanoparticles and CXCL/CXCR targeting strategies has been previously investigated to promote efficient drug delivery to the cancer site [1]. Using a biodegradable cationic heparin-polyethyleneimine (HPEI) nanogel, a recombinant plasmid expressing CXCL10 inhibited the growth of EOC [81]. Additionally, Gao et al. designed a cerium oxide nanoparticle (nanoceria) drug delivery system sensitive to extracellular acidic pH conditions that contained a CXCR4 antagonist and doxorubicin; which enriched targeting to cancer cells and released the chemotherapeutic drug intracellularly [125]. Therefore, the exploration of nanotechnology-based delivery carriers with the combination of CXCL/CXCR strategies may provide promising theranostic strategies to improve the clinical outcomes of EOC. Additionally, machine learning and artificial intelligence based approaches could also be employed to accelerate the discovery and optimization of CXCL/CXCR-targeting therapeutic strategies to improve the clinical outcomes of EOC.

## 4. Conclusions

In this review, the impact of different CXC chemokines and their receptors on EOC progression and their potential theranostic potential were summarized. Evidence suggests that CXC chemokines and their receptors exert a significant influence on the progression of EOC. Due to the complex interactions between these CXCLs/CXCRs and the various non-transformed cell types involved in cancer progression, additional mechanistic studies are required to elucidate the functions of CXCLs/CXCRs accurately and systematically in EOC. This will facilitate mechanism-based approaches for CXCL/CXCR modulation in combination with chemotherapy, targeted therapy, or immunotherapy approaches to improve the prognosis of EOC. 

## Figures and Tables

**Figure 1 ijms-23-02642-f001:**
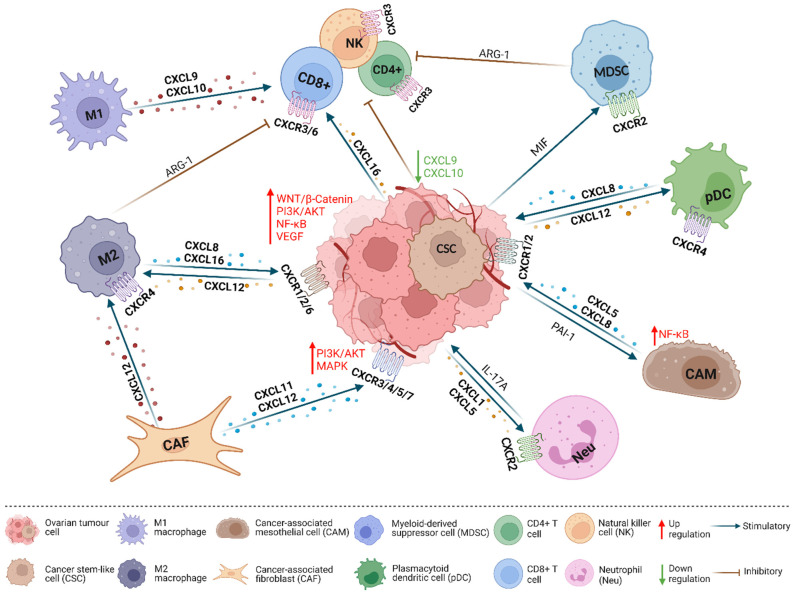
CXCL/CXCR interactions in the tumor microenvironment (TME). Ovarian cancer cells are surrounded by a variety of different types of cells in the TME, and these cells communicate primarily via chemokines to regulate cancer cell proliferation, migration, and angiogenesis. Specifically, CAF-produced CXCL12 acts on CXCR4/7 and leads to activation of the MAPK pathway in cancer cells and the maturation of M2 macrophages which secrete ARG-1 to inhibit anti-tumor immunity. HIF-1α and HIF-2α induce M2 macrophages to express CXCL8/16 in the hypoxic environment; CXCL8/16 activates WNT/β-Catenin, PI3K/AKT, NF-KB, and VEGF signaling pathways in cancer cells and promotes cancer progression. CAF-produced CXCL11 promotes cancer cell proliferation and migration. Cancer cell-secreted CXCL12 recruits pDCs which produce angiogenic factors CXCL8 and TNF-α. Cancer cell-secreted CXCL1/5 recruits Neus, and IL-17A produced by Neus stimulates cancer cells to secrete more CXCL1. Cancer cell-secreted PAI-1 activates the NF-κB pathway and upregulates CXCL5/8 expression in CAMs. CSC-secreted MIF acts on CXCR2 and upregulates ARG-1 in MDSCs. ARG-1, arginase-1; PAI-1, plasminogen activator inhibitor-1; MIF, macrophage migration inhibitory factor. (Created with BioRender.com).

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
