# Peer review of "CXC Chemokine Signaling in Progression of Epithelial Ovarian Cancer: Theranostic Perspectives"

_ijms, 2022, doi:10.3390/ijms23052642_

Round 1
Reviewer 1 Report
In this manuscript, the authors summarized the significance of CXC chemokines/chemokine receptors (CXCR/CXCL), and their roles in regulating the development of epithelial ovarian cancer (EOC). Several review works (Chemokine Receptors in Epithelial Ovarian Cancer, doi: 10.3390/ijms15010361; The emerging role of CXC chemokines in epithelial ovarian cancer, doi.org/10.1530/REP-12-0153) can be found that are similar to the topic discussed herein. The authors should include the discussion of these works and reflect what the present work has different from the ones that have already appeared previously. Accordingly, I would like to recommend a major revision before the manuscript can be accepted.
Point 1:
I recommend the authors to give more discussion on the tumour-suppressor functions of CXCL/CXCR and supplement a summarized illustration.
Point 2:
Since the author also discussed preclinical strategies in section 6, the title ‘CXCL/CXCR-based clinical strategies’ is not appropriate.
Point 3:
Plenty of examples of CXCL/CXCR-based clinical strategies in many types of cancer can be found in section 6, but this review is focused on the EOC and the authors should put more emphasis on the EOC theranostics.
Point 4:
In the biomarker session, the authors need to extend their discussion on these prognostic markers in EOC and corresponding therapeutic strategy and outcomes in preclinical and clinical studies.
Point 5
The authors should put the brief outcomes of these preclinical and clinical studies in Table 2
Point 6
I strongly recommend the authors give some pages to the nanoparticles or carriers with the combination of CXCL/CXCR strategies for EOC theranostics.
Reviewer 2 Report
The review by Huang and colleagues provides an important informational overview of CXC chemokine signaling relevant to the progression of epithelial ovarian cancer. To complement the research-based summary, translational therapeutic information is reviewed as well. This review will be of interest to the cancer research community. The following comments are provided to the authors to strengthen the presentation of the review:
- Grammatical editing is required.
- Following the CXC chemokine and receptor functions in EOC, a Table is presented - - and labeled as Table 2. It should be relabeled and corrected as Table 1.
- Figure 1 is an artistic overview of CXCR/CXCL signaling mechanisms between EOC cells and cells in the TME. It would be helpful to designate the EOC as (orange) and define the abbreviations for each cell type in the legend. Recognizing that abbreviations are defined at the end of the paper, it would help the reader to have the cell types defined earlier in this legend.
- For Table 2 summarizing the CXCRs/CXCLs – associated targets for clinical strategies, a similar suggestion is offered to define the cancer type abbreviations in the legend.
Round 2
Reviewer 1 Report
The manuscript can be accepted in the present form.